# Mixed Tree Nut Snacks Compared to Refined Carbohydrate Snacks Resulted in Weight Loss and Increased Satiety during Both Weight Loss and Weight Maintenance: A 24-Week Randomized Controlled Trial

**DOI:** 10.3390/nu13051512

**Published:** 2021-04-30

**Authors:** Jing Wang, Sijia Wang, Susanne M. Henning, Tianyu Qin, Yajing Pan, Jieping Yang, Jianjun Huang, Chi-Hong Tseng, David Heber, Zhaoping Li

**Affiliations:** 1Center for Human Nutrition, Department of Medicine, David Geffen School of Medicine, University of California Los Angeles, Los Angeles, CA 90095, USA; jingwang023@gmail.com (J.W.); new8090@hotmail.com (S.W.); qintianyu@bucm.edu.cn (T.Q.); panyajing152@sina.com (Y.P.); JiepingYang@mednet.ucla.edu (J.Y.); JianjunHuang@mednet.ucla.edu (J.H.); dheber@mednet.ucla.edu (D.H.); zli@mednet.ucla.edu (Z.L.); 2College of Animal Science and Technology, Hunan Agricultural University, Changsha 410128, China; 3Center of Yangtze River Delta Region Green Pharmaceuticals, College of Pharmaceutical Science & Collaborative Innovation, Zhejiang University of Technology, Hangzhou 310014, China; 4Key Laboratory of Health Cultivation of the Ministry of Education, Beijing University of Chinese Medicine, Beijing 100029, China; 5Department of Statistics Core, David Geffen School of Medicine, University of California Los Angeles, Los Angeles, CA 90095, USA; ctseng@mednet.ucla.edu

**Keywords:** mixed tree nuts, weight loss, weight maintenance, human study, inflammatory markers

## Abstract

Mixed tree nuts (MTNs) are an excellent source of protein and healthy fat contributing to satiety. However, their relatively high caloric content might not be beneficial in a weight loss diet. The present study was designed to test whether including MTNs in a weight loss and maintenance program interferes with weight management compared to a refined carbohydrate pretzel snack (PS). We performed a randomized, controlled, two-arm study in 95 overweight individuals consuming 1.5 oz of MTNs or PS daily as part of a hypocaloric weight loss diet (−500 kcal) over 12 weeks followed by an isocaloric weight maintenance program for 12 weeks. Participants in both groups experienced significant weight loss (12 weeks: −1.6 and −1.9 and 24 weeks: −1.5 and −1.4 kg) compared to baseline in the MTN and PS groups, respectively. However, there was no difference in weight loss and other outcome parameters between the MTN and PS groups. The MTN group showed a significant increase in satiety at 24 weeks. Both groups had a decrease in diastolic blood pressure at 12 weeks. Participants in the MTN group showed significant decreases in heart rate at 4, 12, and 24 weeks. Plasma oleic acid was significantly increased at 12 and 24 weeks in the MTN group but only at 12 weeks in the PS group. Plasma MCP-1 was decreased significantly in the MTN group at 4 weeks. In summary, participants in both groups lost weight, but only the MTN intervention increased satiety at 24 weeks, enhanced retention, decreased heart rate, and increased serum oleic acid at 24 weeks.

## 1. Introduction

Overweight and obesity in the United States and other industrialized and developing countries represent a significant and growing health problem [1,2]. Individuals who are overweight or obese have an increased risk of coronary heart disease (CHD), diabetes (T2D), and metabolic syndrome. Calorie-controlled diets have routinely been demonstrated to help with both short- and long-term weight reduction in individuals who are overweight or obese [3]. It is important for individuals who are following weight loss programs to maintain healthy nutritional patterns and maintain or improve their dietary quality. Nuts might play a significant role as part of a healthy weight loss program and help individuals not only improve their dietary quality but also lower their risk of chronic diseases such as CHD, T2D, and metabolic syndrome. Epidemiologic studies have shown an association between the frequency of nut consumption and body mass index (BMI) and risk of obesity [4,5,6]. Two large, prospective cohort studies (Adventist Health Study and Nurses’ Health Study II) showed significant inverse associations between the frequency of nut consumption and BMI [7,8], whereas no relation was reported in the Physicians’ Health Study [9].

A meta-analysis including nuts in isocaloric diets to replace other food items with high energy density showed a small, non-significant reduction in body weight and BMI between individuals including nuts in their diet compared to regular food [4]. However, the majority of intervention studies were not designed to evaluate body weight change as the primary outcome. In our previous small intervention study, consumption of a daily serving of 53 g of pistachio as part of a hypocaloric diet showed a trend to larger weight loss compared to the consumption of an equal amount of calories from pretzels [10]. Most of the published studies used a single source of nuts instead of a mixture of tree nuts. The diversity in nutrient profiles among the different types of tree nuts may enhance the beneficial effect of tree nuts. Tree nuts have favorable nutrient composition contributing protein, heart-healthy fats, fiber, minerals, vitamins, and phytochemicals to our diet [11,12,13,14,15]. Tree nuts vary in their content of monounsaturated and polyunsaturated fatty acids, fiber, and polyphenols [11]. Therefore, the consumption of a mixture of tree nuts might be superior to the consumption of one type of tree nut. Although the high fat and protein content of nuts might contribute to satiety and the success of a weight loss program, there are concerns that the relatively high caloric content of nuts might not be beneficial in a weight loss diet [16]. Many people in the United States consume ≥2 snacks/day, contributing ∼20–25% of energy intake on average [17,18]. Therefore, it is important to investigate the health outcome of different types of snack food. It was the aim of the present study to determine whether the incorporation of MTNs in a hypocaloric weight loss and weight maintenance diet will lead to weight loss and reduced inflammation compared to the consumption of a hypocaloric diet including equal calories from a carbohydrate source (pretzels).

## 2. Materials and Methods

### 2.1. Study Design

This randomized controlled intervention study was carried out at the Center for Human Nutrition, University of California Los Angeles, CA, USA, in accordance with the guidelines of the Human Subjects Protection Committee of the University of California, Los Angeles. The clinical protocol was approved by the Internal Review Board of the University of California, Los Angeles. All subjects gave written informed consent before the study began. The study was registered in ClinicalTrials.gov under the following identifier: NCT03159689. Participants were recruited through advertisements on social media and local flyers and announcements to previous study participants. The 24-week study was designed as a randomized, controlled, open-label, 2-arm parallel intervention trial with 12 weeks of weight loss followed by 12 weeks of weight maintenance (Figure 1). The study coordinator assigned participants to either consuming MTNs or PS daily based on randomization performed by the statistics core. The randomization algorithm involved sampling probability values from the uniform distribution without a replacement and allocating drawn probabilities to the group assignment. No age or sex stratification was applied. A permuted block design was used with a sampling of 8 per block for a total number of 10 blocks. Enrollment was closed before a target of 154 participants was reached due to stay at home orders from the state of California and closure of the UCLA campus during the COVID-19 pandemic.

Participants in both groups met with a registered dietitian every other week throughout the intervention phase. Participants were provided with personalized hypocaloric (−500 kcal/day) meal plans for the first 12 weeks, followed by isocaloric meal plans for another 12 weeks. Hypocaloric meal plans were estimated based on lean body mass measured by bioimpedance (Tanita-BC418, Tanita Corp, Tokyo, Japan). Meal plans for 1200, 1400, 1600, 1800, or 2000 kcal were prepared to provide the following macronutrient composition: mixed nut group: 30% fat, 15% protein, and 55% carbohydrate, and the PS group was provided calories as follows: 20% fat diet, 15% protein, and 65% carbohydrate. Daily diet checklists were used by individuals to monitor the appropriate consumption of all foods, beverages, and calories each day. A daily snack of 1.5 oz. of MTNs or pretzels with the same caloric content was included in the individual meal plans. Mixed nuts (almonds, cashews, hazelnuts, macadamia, pecans, pistachios, and walnuts) were provided prepackaged in 1.5 oz. portion sizes by the International Tree Nut Council. Packages of mini pretzels were purchased from Snyder’s of Hanover (Hanover, PA, USA). Participants prepared their own food and were asked to record their daily food intake for 3 days prior to the visit to the dietitian. Both meal plans had a similar salt content. Body weight, body mass index, body composition, blood pressure, and satiety were determined at baseline and 4, 12, and 24 weeks. Fasting blood samples were collected at baseline and 4, 12, and 24 weeks to determine plasma total cholesterol, LDL-cholesterol, HDL-cholesterol, triglyceride, inflammatory markers, and red blood cell fatty acids. Participants scored satiety using the hunger–satiety scale from 0 to 7 daily after dinner by the participants for one week at baseline and 4, 12, and 24 weeks. Scores presented are weekly scores, the sum of the 7 daily scores. The scale included the following characteristics: 0 = starving, dizzy, irritable; 1 = very hungry, unable to concentrate; 2 = hunger, ready to eat; 3 = beginning signals of hunger; 4 = comfortable, neither hungry nor full; 5 = comfortably full, satisfied; 6 = very full, feel as if you have overeaten; 7 = stuffed to the point of feeling sick.

### 2.2. Participants

Study participants were healthy free-living women and men with overweight/obesity (BMI: 27.0–35.0 kg/m^2^) and 30–68 years of age. Individuals enrolled in commercial weight loss programs or taking weight loss medication were excluded. In addition, individuals who experienced more than a 5-pound weight gain or loss within 3 months prior to enrollment were excluded from the study. Individuals with thyroid disease, history of chronic disease, following special diets, consuming access alcohol, smoking, or pregnant were excluded. A total of 131 participants were enrolled (nut group: 67; pretzel group: 64). Eleven participants (16.4%) in the nut group and 23 (35.9%; *p* = 0.019) in the pretzel group did not complete the study due to various reasons listed in Appendix A (Figure 1). In general, no adverse events or side effects were observed.

### 2.3. Study Endpoints

The primary endpoint of this intervention study was to determine the effect of daily nut consumption on body weight and composition compared to pretzel intake. Multiple exploratory secondary endpoints included the effect of daily nut consumption on blood pressure, plasma lipids and cholesterol, inflammatory markers, and fatty acids and satiety. All outcome parameters were determined before (baseline) and after 4, 12, and 24 weeks of nut/pretzel consumption.

### 2.4. Weight and Body Composition Measurement

Body weight and body composition were determined at baseline (Week 0) and Weeks 4, 12, and 24. Body composition was determined using the Tanita-BC418 body-fat analyzer (Tanita Corp., Tokyo, Japan).

### 2.5. Plasma Lipids, Cholesterol, and Inflammatory Markers

Fasting plasma total and HDL-cholesterol and triglyceride were analyzed spectrophotometrically using reagents from Pointe Scientific (Pointe Scientific, Canton, MI, USA). Plasma high-sensitivity C-reactive protein (hs-CRP), interleukin 10 (IL-10), tumor necrosis factor-alpha (TNF-α), and monocyte chemoattractant protein-1 (MCP-1) were determined using the HCVD3MAG-67K-01 (EMD Millipore, Billerica, MA, USA) human kit and human cytokine panel HCYTOMAG-60K-06 (EMD Millipore, Billerica, MA, USA) with the Luminex MagPix^®^ analyzer (Luminex, Austin TX, USA).

### 2.6. Plasma Fatty Acid Panel

Plasma fatty acids were extracted with hexane and converted to methyl esters (FAME) according to the method by Bagga et al. [19]. FAME were separated and quantified by use of an Agilent Technologies (San Diego, CA, USA) 5890A Series II Gas Chromatograph fitted with a model 7673 automatic split-injection system and flame ionization detector and an SP2380 stabilized phase fused silica capillary column (30 m × 0.32 mm i.d., 0.25 µm film thickness, Supelco, Inc, Bellefonte, PA, USA). Quantification was based on a standard curve established with a 38 standard mix purchased from NuChek Prep Inc. (Elysian, MN, USA). For quality control, a pooled plasma sample was used. This pooled plasma sample is analyzed with each batch of samples.

### 2.7. Statistical Analyses

The original power calculation was based on the outcome of our previous study [10]. The study demonstrated that the daily consumption of a serving of 53 g of pistachio nuts (about 2 oz.) as part of a hypocaloric diet resulted in a larger weight loss compared to the control group consuming pretzels [10]. We determined that with a sample size of N = 70 per arm and 6 months’ intervention, we will have >80% power to detect a significant difference in weight loss between the mixed nut and pretzel intervention group. Due to early study completion, we performed another power analysis with the final number of study participants, which showed that with 56 subjects in the MTN group and 39 in the PS group, we had 80% power to detect a difference of 1.5 kg in weight change from 0 to 12 weeks between treatment groups. Data were entered into Microsoft Excel and imported into SAS version 9.4. Summary statistics (means and SEM) were generated for the baseline demographic to characterize the study population. Because this is a longitudinal repeated measurement study, a linear mixed-effects model was used to accommodate the within-subject correlation and evaluate the change within the nut and pretzel groups. Within the treatment groups, the percentage change of each outcome between 4, 12, and 24 weeks was compared to baseline. Values of *p* < 0.05 were considered statistically significant. To compare demographic data between the two groups at baseline (Table 1), the Wilcoxon rank-sum test was used to compare continuous variables and Fisher’s exact test to compare categorical variables.

## 3. Results

### 3.1. Participant Flow and Baseline Characteristics

Figure 1 presents the flow diagram of study design, participant randomization, and drop out. There was no significant difference in age, gender, ethnicity, weight, and BMI between participants from both intervention groups at baseline (Table 1).

### 3.2. Compliance, Body Weight, Body Composition, Blood Pressure, and Heart Rate

Both groups were highly compliant with the intervention (nut group: 97.6 ± 0.9%, 95.7 ± 1.6%, and 97.6 ± 1.0% and pretzel group: 97.1 ± 1.5%, 97.6 ± 0.7%, and 98.9 ± 0.4% at 4, 12, and 20 weeks, respectively). The dropout rate was significantly lower in the MTN (16.4%) group compared to the PS (35.9%) group. There was no significant difference in body weight and BMI between the MTN and PS groups (Figure 2 and Appendix A). Participants consuming 1.5 oz. of MTNs or PS daily experienced significant weight loss of −1.6 ± 0.4 and −2.0 ± 0.6 kg of weight from baseline at 12 weeks, respectively, and the MTN group maintained the significant weight loss after another 12 weeks of weight maintenance with an isocaloric diet (Figure 2A). Participants in the PS group experienced a trend to weight increase during the weight maintenance phase (Week 12 compared to 24). The MTN and PS groups also showed a significant decrease in BMI from 31.1 ± 0.4 and 30.7 ± 0.4 kg/m^2^ at baseline to 30.5 ± 0.4 and 29.9 ± 0.5 kg/m^2^ at 12 weeks, respectively (Figure 2C). Diastolic blood pressure was decreased significantly in both groups from 78.5 ± 1.2 mmHG (tree nut) and 76.2 ± 1.2 mmHG (pretzel) at baseline to 76.1 ± 1.2 mmHG (tree nut) and 73.7 ± 1.3 mmHG (pretzel) at 12 weeks compared to baseline, while systolic blood pressure was not changed (Table 2). Heart rate was decreased significantly at Weeks 4, 12, and 24 compared to baseline in participants consuming MTNs, while the heart rate did not change significantly in the PS group at any time point (Table 2).

### 3.3. Plasma Lipids and Fatty Acid Panel

Comparing changes in plasma lipids and the fatty acid panel between the MTN and PS groups did not show any significant differences. At baseline, no difference in fasting plasma triglyceride, total cholesterol, HDL-cholesterol, and plasma fatty acid panel was observed between the MTN and PS groups. There were no changes in fasting plasma triglyceride, total cholesterol, and HDL-cholesterol in both groups comparing 4, 12, and 24 weeks to baseline (Table 2). In the MTN group, plasma oleic acid was significantly increased at 12 and 24 weeks compared to baseline. In the PS group, plasma oleic acid was significantly increased at 12 weeks but not at 24 weeks compared to baseline. In addition, in the PS group, plasma stearic acid was decreased at 12 weeks, and linolenic acid was increased at 24 weeks compared to baseline (Table 3).

### 3.4. Serum Inflammatory Markers

Comparing changes in serum inflammatory markers between the MTN and PS groups did not show any significant differences. At baseline, there was also no difference between serum hs-CRP, IL-10, and TNFα between the MTN and PS groups, while MCP-1 was significantly lower in the pretzel group compared to the MTN group at baseline (Figure 3, Appendix A). Plasma MCP-1 was significantly decreased in the MTN group at 4 weeks compared to baseline, while no significant changes were observed at 12 and 24 weeks (Figure 3B). In addition, there were no significant changes in hs-CRP, IL-10, and TNFα in the MTN and PS groups comparing 4, 12, and 24 weeks to baseline (Figure 3A,C,D).

### 3.5. Satiety Score

At baseline, there was no difference in satiety scores between the MTN and PS groups (Figure 4). At 24 weeks compared to baseline, satiety was increased significantly in the MTN group, while the increase in satiety in the PS group did not reach significance (Figure 4).

## 4. Discussion

Nuts are energy dense, and one might suspect that nut consumption as a snack as part of a hypocaloric diet might hinder weight loss. On the other hand, due to the protein, monounsaturated fat, and micronutrients in tree nuts, consumption may lead to increased satiety and dietary satisfaction. In the present study, no significant difference in weight loss was observed between the MTN and PS groups because participants in both groups experienced significant weight loss. However, tree nuts were associated with increased satiety, enhanced study retention, decreased heart rate, and increased serum oleic acid during weight maintenance.

The present study demonstrated that including 1.5 oz of MTNs daily in a hypocaloric diet for 12 weeks and an isocaloric diet for another 12 weeks did not hinder weight loss and weight maintenance compared to pretzels.

Similar results were found by other investigators using walnut-, almond-, or pistachio-enriched hypocaloric diets [10,20,21]. A study by Rock et al. showed that a walnut-enriched hypocaloric diet promoted weight loss, which was not different compared to that observed with a standard reduced-energy-density diet, with both groups experiencing similar significant weight loss (walnut: −8.5 ± 0.9; control: −7.9 ± 0.6) [20]. A study by Foster et al. also did not find a difference in weight loss between an almond-enriched hypocaloric diet (1200–1500 kcal/d for women and 1500–1800 kcal/d for men) including 56 g of almonds per day compared to a nut-free hypocaloric diet in 123 participants [21]. Furthermore, our previous randomized controlled intervention study with 50 participants [10] also did not lead to significantly greater weight loss with 53 g of pistachio compared to 56 g pretzels daily in a 500 kcal hypocaloric weight loss diet for 12 weeks (pistachio: −3.7 ± 1.4 kg; pretzel: −2.7 ± 2.2 kg).

On the other hand, investigators studying an almond-supplemented hypocaloric diet (50 g/day) demonstrated greater weight loss compared to a nut-free diet, where calories were supplied by meat or fat instead of nuts [22]. The hypocaloric almond diet led to a greater weight loss of −3.7 ± 2.8 in comparison to the hypocaloric nut-free diet (−1.3 ± 3.6) [22].

Together, these studies indicate that including nuts in a weight loss intervention does not hinder and may support weight loss. Whether significantly greater weight loss is observed in the nut group appears to depend on the diet composition in the control group and possibly on the macronutrient composition of the diets.

Although there was no difference in satiety between the MTN and PS groups, the MTN group experienced an increase in satiety at 24 weeks, and more participants consuming the tree nut snack completed the study. In the present study, there was a significantly higher dropout rate in the pretzel group (35.9%) compared to the tree nut group (13.4%) in spite of matching retention efforts.

Increases in satiety have been demonstrated in other nut studies [23,24,25]. Potential mechanisms whereby nut consumption increases satiety [24,25] have been suggested to be based on a decrease in appetite-related hormones ghrelin and leptin [24]. However, a study by Brennan et al. did not find a change in satiety-mediating hormones in an acute study of a walnut-containing diet demonstrated increased pre-lunch satiety and fullness ratings compared to a placebo diet [25]. A study by Sayer et al. compared the postprandial effect of almond with isoenergetic baked goods on postprandial hunger, desire to eat, fullness, and neural response to visual food stimuli via functional magnetic resonance scans [26]. Satiety was affected by energy and macronutrient content and not by any unique almond variables in that study. In summary, data from our study and others demonstrate that the consumption of a tree nut snack increases satiety due to their increased energy density.

Several studies have also investigated the effect of hypocaloric nut-containing diets on cardiovascular disease risk factors [27,28]. In the present study, no changes were observed in fasting triglyceride, total cholesterol, HDL-cholesterol, hs-CRP, IL-10, and TNFα. However, in the present study, heart rate was decreased at all time points compared to baseline in the tree nut group, and diastolic blood pressure was decreased after 12 weeks of tree nut consumption, suggesting cardiovascular benefits from nuts independent of weight loss. In the other studies mentioned above, when weight loss ranged between 5 and 8 kg in 6 months, a significant decrease in cardiovascular markers was observed [20,21,22]. Participants in the walnut group in the study by Rock et al. also experienced a decrease in heart rate during a step test [20]. A decrease in heart rate might be related to the increase in endothelial function observed in a study by Dikariyanto et al. in individuals consuming almonds as part of an isocaloric diet (20% of caloric intake) for 8 weeks possibly due to their high arginine content which could stimulate nitric oxide production by endothelial cells [15,29].

In the tree nut family, almond, cashew, hazelnut, macadamia, pecan, and pistachio nuts have the highest content of monounsaturated fatty acids, while walnuts are rich in linolenic acid [11,30]. In the present study, a significant increase in plasma oleic acid was found at 12 and 24 weeks of MTN intake and at 12 weeks of PS intake. Two other studies investigating the addition of either cashew or almonds to an isocaloric diet showed an increase in plasma oleic acid [31,32]. A study by Nishi et al. demonstrated that the increase in plasma oleic acid was associated with a decrease in cardiovascular risk, as calculated using Framingham’s 10-year risk score [31]. In the present study, we observed an increase in plasma oleic acid, and significantly improved diastolic blood pressure and heart rate.

Study limitations included a lower number of participants than planned due to the COVID pandemic. With 95 subjects, we have 80% power to detect a difference of 1.5 kg in weight change from 0 to 12 weeks between treatment groups. With this power, we did not see any weight loss difference between the groups. The other limitation is the significantly higher dropout rate in the pretzel control group compared to the mixed tree nut snack. The choice of another control snack might reduce drop out and might demonstrate a difference in weight loss between the tree nut and the control group. In addition, we matched the calorie of the two snacks but not the macronutrient composition. Because the study was performed in free-living participants who provided their own food, we do not have data on their background diet.

## 5. Conclusions

No difference in weight loss was observed between the MTN and PS groups because participants in both groups experienced significant weight loss during the 12-week weight loss program. Only participants in the MTN group maintained a lower weight, while the PS group experienced a trend to weight increase in the 12-week maintenance phase. Increases in serum oleic acid and a decrease in heart rate following tree nut consumption indicate that there may be other heart health benefits of tree nut consumption as part of a hypocaloric or weight maintenance diet.

## Figures and Tables

**Figure 1 nutrients-13-01512-f001:**
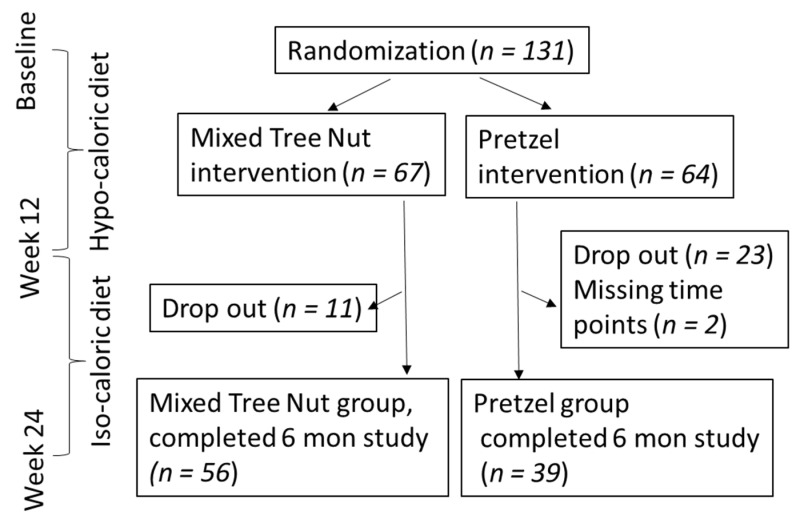
Study flow diagram for the randomized controlled trial of a mixed tree nut supplement compared to pretzel supplement among healthy adults with overweight/obesity consuming a hypocaloric followed by an isocaloric diet.

**Figure 2 nutrients-13-01512-f002:**
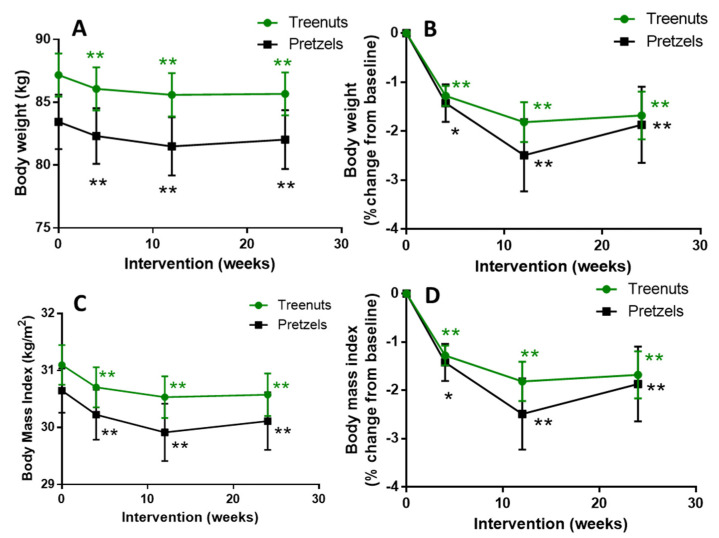
Effect of tree nuts and pretzels on body weight (**A**,**B**) and BMI (**C**,**D**) expressed as absolute values and percent change from baseline. Data are means ± SEM. * *p* < 0.05, ** *p* < 0.001 compared to baseline in the same treatment group. A linear mixed-effects model was used to analyze the repeated measurements within subjects and evaluate the change within and between the tree nut and pretzel control groups. Tree nut, *N* = 56; pretzel *N* = 39.

**Figure 3 nutrients-13-01512-f003:**
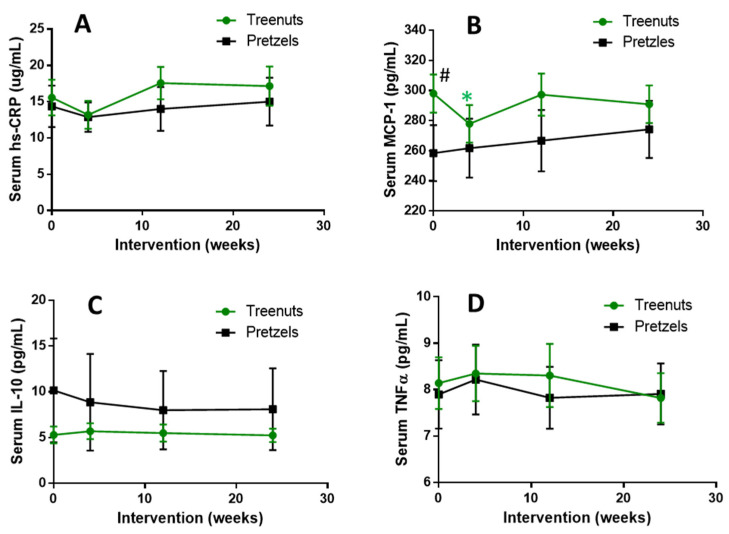
Effect of tree nuts and pretzels on (**A**) serum hs-CRP, (**B**) serum MCP-1, (**C**) serum IL-10, and (**D**) TNF-α concentrations. Data are means ± SEM. * *p* < 0.05, compared to baseline in the same treatment group. # *p* < 0.05, comparing the tree nut and pretzel groups. Tree nut, *N* = 56; pretzel *N* = 39. Hs-CRP, high-sensitivity C-reactive protein; MCP-1, monocyte chemoattractant protein-1; IL-10, interleukin 10 and TNFα, tumor necrosis factor-alpha. A linear mixed-effects model was used to analyze the repeated measurements within subjects and evaluate the change within and between the tree nut and pretzel control groups.

**Figure 4 nutrients-13-01512-f004:**
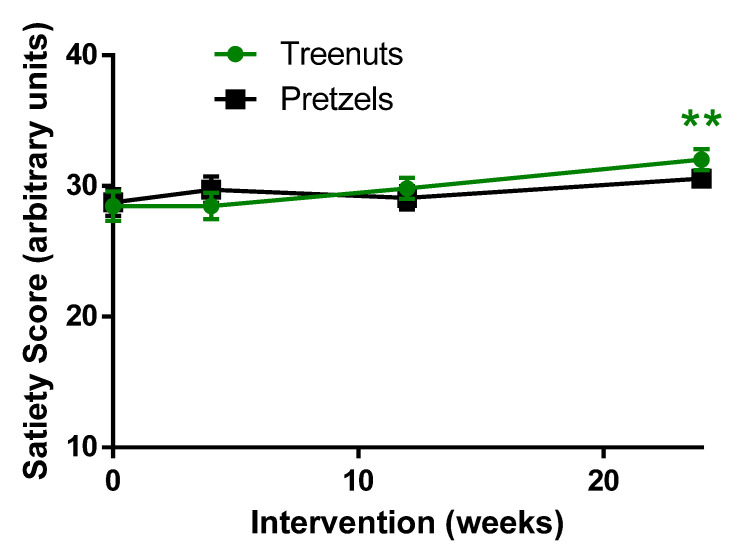
Effect of tree nuts and pretzels on satiety. Data are means ± SEM. ** *p* < 0.0001 compared to baseline in the same treatment group. Tree nut, *N* = 56; pretzel *N* = 39. A linear mixed-effects model was used to analyze the repeated measurements within subjects and evaluate the change within and between the tree nut and pretzel control groups.

**Table 1 nutrients-13-01512-t001:** Baseline characteristics of participants who completed the study.

	Tree Nut (*N* = 56)	Pretzel (*N* = 39)	*p*-Value
Age (years)	48.3 ± 1.9	46.9 ± 1.7	0.8115
Gender (F/M)	39/17	32/7	0.2313
Race/Ethnicity (Anglo/AA/Asian/Native Hawaiian or other Pacific Islander/other) (%)	35/12/7/1/1 (63/21/12/2/2)	24/9/5/1/0 (62/23/13/2/0)	0.948
Weight (kg)	87.2 ± 1.7	83.4 ± 2.2	0.1674
BMI (kg/m^2^)	31.1 ± 0.4	30.7 ± 0.4	0.3842

Data are means ± SEM; *p*-value comparing the tree nut and pretzel groups. The Wilcoxon rank-sum test was used to compare continuous variables and Fisher’s exact test to compare categorical variables.

**Table 2 nutrients-13-01512-t002:** Effect of tree nut and pretzel consumption on fasting plasma triglyceride, total cholesterol, HDL-cholesterol, and diastolic and systolic blood pressure.

	Tree Nut				Pretzel			
	BL	4	12	24	BL	4	12	24
Triglyceride (mg/dL)	88 ± 4.8	88 ± 4.2	94 ± 5.9	85 ± 4.8	84 ± 6.8	79 ± 5.0	84 ± 5.3	84 ± 5.8
Total Chol (mg/dL)	169 ± 6.0	168 ± 6.0	169 ± 6.2	163 ± 5.6	172 ± 8.5	169 ± 8.5	174 ± 7.1	169 ± 7.8
HDL-Chol (mg/dL)	33 ± 1.2	33 ± 1.2	34 ± 1.2	33 ± 1.1	31 ± 1.5	30 ± 1.5	32 ± 1.5	32 ± 1.8
Diastolic BP (mmHG)	78 ± 1.2	77 ± 1.4	76 ± 1.2 *	77 ± 1.3	76 ± 1.1	76 ± 1.2	74 ± 1.4 *	74 ± 1.5
Systolic BP (mmHG)	123 ± 1.6	121 ± 1.9	121 ± 1.8	121 ± 1.8	121 ± 2	120 ± 2.3	120 ± 2.3	118 ± 1.9
Heart Rate (BPM)	74 ± 1.6	71 ± 1.5 *	71 ± 1.3 **	71 ± 1.3 *	73 ± 1.6	71 ± 1.7	71 ± 1.7	73 ± 1.5

Data are means ± SEM; * *p*-value (0.05) ** *p*-value (0.01) comparing each time point to baseline separately in the tree nut and pretzel group. Tree nut, *N* = 56; pretzel, *N* = 39. A linear mixed-effects model was used to analyze the repeated measurements within subjects and evaluate the change within and between the tree nut and pretzel control groups. HDL-Chol, high-density lipoprotein cholesterol, BP blood pressure, BPM beats per minute.

**Table 3 nutrients-13-01512-t003:** Effect of tree nut and pretzel consumption on plasma fatty acids.

Fatty		Tree Nut			Pretzel	
Acids ^#^	0	12	24	0	12	24
Myristic	0.3 ± 0.4	0.3 ± 0.5	0.3 ± 0.4	0.3 ± 0.5	0.3 ± 0.5	0.4 ± 0.5
Palmitic	22.1 ± 1.8	22.0 ± 1.8	21.8 ± 1.5	22.3 ± 1.9	22.6 ± 2.1	22.6 ± 2.2
Stearic	7.1 ± 0.6	7.0 ± 0.5	7.0 ± 0.7	6.9 ± 0.6	6.7 ± 0.7 *	6.9 ± 0.7
Oleic	20.8 ± 2.6	21.6 ± 2.4 *	21.7 ± 2.7 *	21.2 ± 2.9	21.7 ± 2.7 *	21.2 ± 2.8
Linoleic	35.6 ± 4.0	35.3 ± 4.0	35.2 ± 4.1	34.8 ± 3.8	34.3 ± 3.8	34.7 ± 4.1
Linolenic	0.6 ± 0.3	0.6 ± 0.3	0.6 ± 0.3	0.6 ± 0.3	0.6 ± 0.3	0.7 ± 0.2 *
Arachidonic	8.8 ± 2.3	8.6 ± 2.3	8.6 ± 2.1	9.0 ± 2.6	8.8 ± 2.3	8.6 ± 2.3
EPA	0.7 ± 0.5	0.6 ± 0.4	0.7 ± 0.4	0.7 ± 0.5	0.6 ± 0.3	0.6 ± 0.3
DHA	1.7 ± 0.7	1.6 ± 0.6	1.7 ± 0.6	1.6 ± 0.6	1.7 ± 0.7	1.7 ± 0.8

Data are means ± SEM. * *p* ≤ 0.05 compared to baseline. # Plasma fatty acid concentration is expressed as a percent of total lipids. EPA, eicosapentaenoic acid; DHA, docosahexaenoic acid. Tree nut, *N* = 56; pretzel *N* = 39. A linear mixed-effects model was used to analyze the repeated measurements within subjects and evaluate the change within and between the tree nut and pretzel control groups.

## Data Availability

Data described in the manuscript will be made available by the corresponding author upon request pending application and approval.

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
