# Peer review of "Mixed Tree Nut Snacks Compared to Refined Carbohydrate Snacks Resulted in Weight Loss and Increased Satiety during Both Weight Loss and Weight Maintenance: A 24-Week Randomized Controlled Trial"

_nutrients, 2021, doi:10.3390/nu13051512_

Round 1

Reviewer 1 Report

Wang et al. reported the comparison between the effect of mixed tree nuts and pretzels on weight loss and maintenance, metabolic outcomes, and satiety. Although the topic is potentially important and interesting, there are many concerns, which should be sufficiently addressed to improve the quality of the manuscript.

1) The authors conducted a comparison between baseline and intervention period within each of the snack groups and this is one of the major limitations of this study. According to the present title and aim, they should compare the effects of intervention between snacks by difference-in-difference design (additionally to the present analysis) and mainly interpret the results derived from the re-analysis.

2) High incompleteness in the pretzel group (35.9%) is another limitation in the present study. This may lead to the absence of significant associations for some outcomes in the pretzel group due to its lower statistical power. Please provide a more detailed explanation of the reason for dropout, instead of “personal reason”. Additionally, the difference between participants who completed and incompleted intervention should be compared and the influence on the present findings should be discussed in the limitation section.

Introduction:

3) Several references were only mentioned in the discussion section. Previous studies which examined the effect of nut intake on weight loss and maintenance should be referred to in the introduction section. This is the same as the references only mentioned in the introduction section but not the discussion section.

Methods

4) Study design

Please describe the information on recruitment procedure, inclusion and exclusion criteria, randomization methods (e.g., age-and sex-stratified randomization), and blinding of intervention.

5) Line 155-160

The number of participants is lower than 140, because of the earlier closing of enrollment and a relatively high dropout rate. Please re-estimate the statistical power according to the actual number of participants.

6) Line 161-166

Please state the alpha level of the statistical significance. Further, please provide the statistical methods used to compare the values in Table 1. Although the authors described that the percentage change of each outcome was compared within nuts/pretzel groups, only Figure 1 shows such values. I recommend the authors show all outcomes using the percentage change or absolute values. At least, they should precisely describe how to conduct statistical analysis.

Results

7) Tables and Figures

Data provided in Figures 2 and 3 should be provided in Tables because Tables are more informative than Figures and some Tables may be combined according to this revision. Data provided in Figures 2A and 2B should be checked and the values of plasma lipids on 4wks should be shown as well as other outcomes. Moreover, each of the Tables should be self-explanatory; the methods of statistical analysis conducted should be provided in footnotes.

8) Discussion

Because the authors did not compare the effects of intervention between snacks, the present discussion is not appropriate. Further, most of the descriptions are just a review of the previous studies. They should interpret the findings derived from re-analysis using the findings from the previous studies (i.e., explain the mechanism behind the similarity or difference between the present and the previous studies). They also should discuss the strength and limitation of the present study.

9) Line 279-281

I don’t understand why the authors think so. Please provide a sufficient rationale for this sentence.

10) Conclusion

The conclusion is also inappropriate as well as the discussion. Information in line 322-324 should be provided in the introduction.

Author Response

We would like to thank the Reviewer for the detailed and thoughtful comments. After responding to the comments and making revisions we feel the manuscript is significantly improved. We attached a point by point responses to the Reviewer’s comments.

Reviewer 2 Report

The present RCT assessed whether including mixed tree nuts in a weight loss program impacts on weight management compared to a snack based on refined carbohydrate (pretzel).

The study was interesting, clear, and well-written.

However, I have several major concerns:

  • The macronutrient composition of the diets of the two groups was different. The two groups should have received different nutritional schemes in order to pair the macronutrient composition. This way, it would have been possible to truly discriminate the effects of each snack itself. This should be recognized as a limitation and discussed.
  • The short follow-up of the RCT
  • The high (and significantly different) rates of drop-out from the two arms of the trial
  • The likely lack of power of the trial owing to the high drop-out rate (95 individuals were enrolled instead of the required 140)
  • A section dealing with the limitations of the study was not present in the paper.

Author Response

(The authors gave the same response as above.)

Round 2

Reviewer 2 Report

The limitations of the study must be clearly stated. Firts of all, the fact that the present trial didn' t reach an adequate power.

A 6-mo study duration of the study is not a strenght of the study.

The claims relative to the study findings in the Discussion should be greatly attenuated, not being supported by the results. 

Author Response

4.14.2021

Reviewer 2_ Second Round Comments

Below please find the point by point responses to the comments:

The limitations of the study must be clearly stated. First of all, the fact that the present trial didn' t reach an adequate power.

Response: With the reduced number of study participants (due to early completion because of the COVID pandemic) we had 80% power to detect a difference of 1.5 kg in weight change from 0-12 weeks between treatment groups. With this power, we did not see any difference in weight loss between the mixed tree nut and pretzel group. We added a sentence to the statistical section about the new power calculation and a sentence to the limitation section in the discussion.

A 6-mo study duration of the study is not a strength of the study.

Response: We removed the sentence about the strength.

The claims relative to the study findings in the Discussion should be greatly attenuated, not being supported by the results. 

Response: We revised the second paragraph in the discussion to reflect the results of the study.